# Export of Environment Goods from China, Importing Countries' Energy Mix, and Carbon Emission Intensity

Weidong Huo [1], Xiaoxian Chen [1,*] and Yacheng Zhou [2]

1   School of Finance and Trade, Liaoning University, Shenyang 110036, China; huoweidong@lnu.edu.cn
2   School of International Business, Southwestern University of Finance and Economics, Chengdu 611130, China; zhouyacheng507024293@gmail.com
*   Correspondence: 4022110030@smail.lnu.edu.cn; Tel.: +86-155-2407-0958

**Abstract:** Facing the rising global temperature, China, the largest annual carbon emitter, is constantly fulfilling its obligations and acting to inject Chinese impetus into global climate action. Under this background, this paper uses an IV-TSLS regression model to empirically explore the impact of China's Environment Goods Export (CEGE) on the Carbon Emission Intensity of Importing Countries (CEIIC), including a sample of 187 countries, covering the period from 2012 to 2020. We find that the CEGE can reduce the CEIIC by changing the energy mix of importing countries. All five categories of environment goods classified by their protection functions can significantly reduce the CEIIC. Among them, the goods used for the disposal and recycling of waste and pollutants, emission monitoring, and renewable energy projects have the most powerful inhibition effect. The inhibition effect of the CEGE on the Carbon Emission Intensity (CEI) in South America is the greatest, followed by Africa, Oceania, and Asia, while this effect is insignificant in European and North American countries. The CEGE has a stronger inhibition effect on the CEI of non-APEC countries than APEC countries. The CEGE has a far greater inhibition effect on the CEI of six economic corridor countries than the other countries.

**Keywords:** environment goods export; energy mix; carbon emission intensity

## 1. Introduction

Earth experienced its five warmest years on record during 2015–2019, and the concentrations of greenhouse gases reached a new record in 2019 and are increasing, which will fuel future global heat. Climate change is bringing many negative impacts on human production and life. For example, extreme weather is frequent, the retreating of polar glaciers is accelerating sea-level rise, and the increasing ocean acidification has been affecting the survival of marine life. Although global economic growth slowed, and greenhouse gas emissions decreased by about 6% in 2020 due to the outbreak of COVID-19 [1], this case is only temporary, and climate change continues afterward. For this reason, human beings have never stopped paying attention to the environment and climate. From the first United Nations Conference with the theme of environmental issues held in Stockholm in 1972 to the "Stockholm+50" conference in 2022, humanity has paid attention to environmental and sustainable development issues for half a century. At the "Stockholm+50" conference, leaders of various countries called on all mankind to jointly undertake the mission and responsibility of protecting the environment and responding to climate change [2].

The reason for rising global temperature is that greenhouse gas emissions have exceeded the amount that nature can neutralize. To stop global temperatures from continuing to rise, carbon dioxide emissions must decrease considerably. China, with the largest carbon emissions, has been actively participating in global environmental governance and climate action from its beginning. Since Premier Zhou Enlai attended the first United Nations Conference on the Human Environment in 1972, China has never been absent from

any large-scale international conference on this topic. At the 75th United Nations General Assembly in 2020, President Xi Jinping solemnly announced that China will reach a peak in carbon dioxide emissions by 2030 and achieve carbon neutrality by 2060. This is China's solemn commitment to address global warming. In addition to reducing domestic carbon emissions, China should actively help other countries around the world to reduce carbon dioxide emissions through green trade, green investment, and participating in multilateral dialogue mechanisms that focus on environmental issues, which will make important contributions to global climate action. Hence, this paper focuses on whether China has contributed to global climate governance through green trade.

What is green trade? From the viewpoints of academic circles and relevant policy documents, "green trade" has two types of connotations: narrow and broad. In a narrow sense, green trade can be summarized as the trade of products (or services) that meet certain environmental standards or can serve to manage and improve the ecological environment. In a broad sense, green trade not only covers the above concepts but also includes the institutional systems related to the development of green trade, such as relevant policies and safeguard mechanisms. This article focuses on the trade of environment goods in the narrow sense of green trade, that is, the trade of products that can serve the governance and improvement of the ecological environment. Currently, the literature has no unified definition for "environment goods". This article defines environment goods as those listed in the "APEC LIST OF ENVIRONMENT GOODS" issued by the Asia-Pacific Economic Cooperation (APEC) in 2012. The list of environment goods contains a total of 54 types, mainly used for the exploitation of renewable energy, treatment of waste and pollutants, and monitoring of emissions in the production process. Therefore, vigorously developing trade in environment goods is a specific measure in the field of trade to protect the ecological environment and deal with climate change. Based on this description, this paper tries to check whether China, as an APEC member, has made a positive contribution to reducing the Carbon Emission Intensity of Importing Countries (CEIIC) by China's Environment Goods Export (CEGE). We include a sample of 187 countries covering the period from 2012 to 2020. This case raises two questions: (1) what are the intrinsic mechanisms of influence? and (2) are there heterogeneous impacts at the product and country levels? Answering these questions can provide a concrete starting point for China to participate in climate governance, which can contribute to the realization of the landmark 2030 Agenda.

The problem for this study actually belongs to the category of the effects of trade on the environment. The current research on the effects of trade on the environment has five categories. The first category explores the environmental effects of overall trade from three aspects: scale effect, technology effect, and composition effect. Grossman and Krueger (1995) [3] proposed this analytical idea for the first time, later adopted by other scholars [4–7]. In 2009, the World Trade Organization and the United Nations Environment Programme took the above three effects as the theoretical basis for analyzing the effects of trade on the environment. The second category is the literature represented by Frankel and Rose (2002) [8] and Frankel and Romer (1999) [9], which construct the Instrumental Variable (IV) of trade and explore the effects of foreign trade on the environment. The third category of the literature mainly uses the input-output method to measure the embodied carbon in trade to learn the impact of trade on the pattern and evolution of carbon dioxide emissions. The specific calculation methods are mainly the Single Region Input-Output (SRIO) model [10–14] and Multi-Regional Input-Output (MRIO) models [15–19]. The fourth category of the literature mainly studies the impact of trade liberalization on carbon emissions [20,21], including the impact of environment goods trade liberalization on carbon emissions [22]. The fifth category of the literature investigates the effects of trade in certain types of products on environment, and this kind of literature is relatively rare [23,24]. In particular, Mao et al. (2022) [25] innovatively explored the effects of Chinese environment goods trade on domestic carbon dioxide emissions. Alvi et al. (2023) examined the impact of environmental and non-environmental goods trade on carbon emissions in high and middle-income countries [26].

This paper has four innovations as follows. First, this study pays more attention to trade in environment goods and explores its impact on the CEIIC, while most of the existing literature explores the impact of overall trade on the environment from three aspects: scale effect, technology effect, and structure effect. Second, Mao et al. (2022) [25] mainly explored the impact of China's environment goods trade on domestic carbon emissions. In contrast, this paper considers environmental issues from the perspective of the "global village" and explores the effects of trade in environment goods on the Carbon Emission Intensity (CEI) of countries around the world. Third, starting from the level of environment goods, this paper deeply analyzes the environmental protection functions of various environment goods, tries to find the theoretical mechanism for reducing the CEI, and conducts empirical tests, which have been ignored in previous literature. Fourth, this study recognizes the importance of clean and renewable energy exploitation for climate governance. According to the findings, environment goods play a significant role in promoting the transformation of the energy mix to achieve the goal of CEI reduction in an effective way, which are the innovative arguments and conclusions of this paper.

The rest of this paper is arranged as follows: Section 2 sets out the theoretical mechanisms and research hypotheses, which discusses the internal mechanism of CEGE affecting the CEIIC from the perspective of the environmental protection function of products. Section 3 introduces the model setting, data resources, and data processing methods. Section 4 presents and analyzes the regression results. Section 5 summarizes the research conclusions and extracts valuable countermeasures and suggestions based on the research conclusions.

## 2. Mechanisms and Hypotheses

### 2.1. Impact of CEGE on the CEIIC

Overall, previous studies on the effects of trade on the environment investigate the mechanism analysis from three aspects, including scale effect, technology effect, and composition effect [6]. However, environment goods trade, compared with the overall trade, has the characteristics of a small trade scale, fewer product types, and less impact scope, thereby needing examination. Thus, the impact mechanism of environment goods trade on the environment is different from the literature. From the perspective of the environmental protection functions of environment goods, this paper tries to find out the possible impact mechanism of environment goods on CEI. Table 1 represents the environmental protection functions of 54 environment goods.

Table 1 shows that all environment goods have five categories according to their environmental protection functions in Table 2. The first category of goods contains only one product, which is bamboo flooring. As an environmental protection function, this production can save water resources. The second category of goods is biomass-fired boilers and their parts. Biomass fuels are actually clean and renewable energy with less carbon dioxide emissions. Hence, their environmental protection functions are to reduce air pollution in energy consumption. The third category of goods is products related to the exploitation of clean and renewable energy, which mainly includes solar, wind, water, and other generator sets and their components. These goods can replace traditional energy for power generation, which can greatly reduce carbon dioxide emissions. The fourth category mainly includes equipment and components used to treat waste and pollutants, such as mixers, flocculators, etc., for solid waste treatment. These goods can decrease the detrimental impact of industrial production on the environment. The fifth category is mainly used to monitor the emissions in the production process, such as detection instruments of air, water quality, and air pollutant emissions. Although these goods cannot directly reduce carbon dioxide emissions and environmental pollution, they can serve as a warning, which encourages producers to reduce emissions of greenhouse gases and pollutants. According to the environmental protection functions of various products, environment goods can effectively reduce carbon dioxide emissions, thereby reducing CEI. Based on this analysis, this study proposes Hypothesis 1.

**Table 1.** Description of environmental protection functions of environment goods.

| HS Code | Description | HS Code | Description |
|---|---|---|---|
| 441872 | Save water and other resources | 850300 | Renewable energy (wind, etc.) power generation |
| 840290 | Reduce pollution by using biomass fuel | 850490 | Renewable energy (wind, etc.) power generation |
| 840410 | Reduce pollution by using biomass fuel | 851410 | Handle pollutants |
| 840420 | Reduce pollution by using biomass fuel | 851420 | Handle pollutants |
| 840490 | Reduce pollution by using biomass fuel | 851430 | Handle pollutants |
| 840690 | Efficient production of geothermal energy (renewable energy) | 851490 | Handle pollutants |
| 841182 | Renewable energy (biogas, etc.) power generation | 854140 | Renewable energy (solar) power generation |
| 841199 | Renewable energy (biogas, etc.) power generation | 854390 | Kill harmful microorganisms |
| 841290 | Converting wind energy into kinetic energy by wind turbines | 901380 | Renewable energy (solar) power generation |
| 841780 | Destroy pollutants | 901390 | Renewable energy (solar) power generation |
| 841790 | Destroy pollutants | 901580 | Monitoring and preventing natural disaster risks |
| 841919 | Renewable energy (solar) heating water | 902610 | Emission monitoring |
| 841939 | Treatment of sludge and wastewater | 902620 | Emission monitoring |
| 841960 | Help to remove contaminants through condensation | 902680 | Emission monitoring |
| 841989 | remove contaminants | 902690 | Emission monitoring |
| 841990 | Renewable energy (solar) heating water | 902710 | Emission monitoring |
| 842121 | Filter pollutants in wastewater | 902720 | Emission monitoring |
| 842129 | Filter pollutants in wastewater | 902730 | Emission monitoring |
| 842139 | Handle pollutants in gas | 902750 | Emission monitoring |
| 842199 | Handle pollutants | 902780 | Emission monitoring |
| 847420 | Handle solid waste | 902790 | Emission monitoring |
| 847982 | Handle waste | 903149 | Emission monitoring |
| 847989 | Handle waste | 903180 | Emission monitoring |
| 847990 | Handle waste | 903190 | Emission monitoring |
| 850164 | Renewable energy (biomass fuel) power generation | 903289 | Emission monitoring |
| 850231 | Renewable energy (wind) power generation | 903290 | Emission monitoring |
| 850239 | Renewable energy (biogas, etc.) for power generation | 903300 | Help machines in Chapter 90 come into play |

**Table 2.** Goods classification according to their environmental functions.

| No. | HS Code | Main Environmental Protection Functions |
|---|---|---|
| 1 | 441872 | ① Save water and other resources |
| 2 | 840290, 840410, 840420, 840490 | ② Use biomass fuel to reduce pollution |
| 3 | 840690, 841182, 841199, 850164, 850231, 850239, 850300, 841290, 841919, 841990, 850490, 854140, 901380, 901390 | ③ Used for the exploitation of clean and renewable energy |
| 4 | 847420, 847982, 847989, 847990, 841780, 841790, 841939, 841960, 841989, 851410, 851420, 851430, 851490, 842121, 842129, 842139, 842199, 854390 | ④ Used for the disposal and recycling of waste, pollutants, etc. |
| 5 | 901580, 902610, 902620, 902680, 902690, 902710, 902720, 902730, 902750, 902780, 902790, 903149, 903180, 903190, 903289, 903290, 903300 | ⑤ Used for environmental and pollution monitoring |

**Hypothesis 1.** *CEGE can significantly decrease the CEI of importing countries.*

According to the above analysis, the second and third categories of environment goods can change the energy mix of importing countries by increasing the consumption of clean and renewable energy, which can promote CEI reduction. This effect is the most

critical impact mechanism of CEGE on CEIIC. Specifically, the main reason for high carbon dioxide emissions in existing production and life is that a large amount of traditional energy (such as coal) is used for power generation and heating. However, among the above environment goods, the second category of goods directly uses biomass fuels, a clean and renewable energy source, to heat boilers. The combustion of biomass fuels releases much less carbon dioxide than conventional fuels. The third category of goods uses renewable energies like solar, wind, hydro, and biomass energies to generate electricity, which is an environmentally friendly alternative to the traditional fossil energy power generation mode. The power generation mode using clean and renewable energy produces almost no carbon dioxide emissions. Based on the above analysis, this study proposes Hypothesis 2. However, the CEIIC can be reduced by CEGE through the above mechanism only if the above environment goods are widely used by importing countries.

**Hypothesis 2.** *CEGE increases the proportion of clean and renewable energy used or used for power generation in importing countries, thereby reducing its CEI.*

*2.2. Heterogeneous Impact of CEGE on the CEIIC*

Based on the mentioned analysis, goods in categories 2 and 3 can increase the consumption of clean and renewable energy, thereby effectively reducing CEI. However, the ambiguity is whether and how goods in categories 1, 4, and 5 affect CEI. Hence, this paper proposes Hypothesis 3.

**Hypothesis 3.** *The goods in categories 2 and 3 can reduce the CEIIC more effectively than any other category.*

Chinese environmental products have different functions in various importing countries due to the differences in the maturity and technical level of industries related to environment goods in different countries. For example, industries related to the environment are the dominant industries in developed economies like Europe and North America, which have the most advanced production and cleanliness techniques. Therefore, the CEGE has a limited impact on CEI in developed economies. On the contrary, for the backward economies of South America and Africa, China's export of environment goods, especially renewable energy power generation goods, which are China's dominant goods, may be able to greatly change the existing energy mix of these countries, thereby reducing CEI. For example, the hydropower project undertaken by China International Water and Electric CORP. (Cwe) provides 1.039 billion kWh of electricity for the local area every year in Isimba, Uganda, and increases the capacity of power generation by as much as 20%. In addition, according to the "One Belt, One Road" Eco-Environmental Big Data Analysis Report 2020, one of the three 100 GW-level renewable energy development projects is located in Africa. The two continents of Oceania and Asia are the key regions for China's cooperation in renewable energy projects. Hence, China's export of environment goods to these countries may effectively change their energy mix. Based on this analysis, this study proposes Hypothesis 4. However, the premise of Hypothesis 4 is that environment goods are widely used to generate power by importing countries.

**Hypothesis 4.** *CEGE has the strongest inhibitory effect on CEI in South America and Africa, followed by Oceania and Asia, while this effect is insignificant in Europe and North America.*

APEC countries are the initial signatories of the list of environment goods, so the development level of the related industries is relatively advanced. Therefore, China's environment goods may lack a comparative advantage over other APEC members, which makes it difficult for CEGE to impact the domestic energy mix and CEI in these countries significantly. Based on this analysis, this study proposes Hypothesis 5.

**Hypothesis 5.** *CEGE has a stronger inhibitory effect on the CEI of non-APEC countries than APEC countries.*

Driven by the "One Belt, One Road" platform, there are more and more cases of Chinese energy companies "going-out". It has undertaken many renewable energy projects, most of which are located in the Six Economic Corridors (SEC). According to the statistics of the Big Data Analysis 2020, two of the three 100 GW renewable energy projects are located in the New Eurasian Land Bridge Economic Corridor and the Bangladesh-China-India-Myanmar Economic Corridor. In addition, the China-Pakistan Economic Corridor is the earliest partner for China's renewable energy "going-out". The China-Indochina Peninsula has also developed into an important destination for Chinese energy companies' investment overseas. The main investment fields are hydropower and photovoltaic power generation. In this way, the energy mix of SEC countries will change, and their CEI will decrease. According to this analysis, this study proposes Hypothesis 6. Analogously, the premise of Hypothesis 6 is also that environment goods are widely used to generate power by importing countries.

**Hypothesis 6.** *CEGE has a stronger inhibitory effect on the CEI of SEC countries than the other countries.*

All the hypotheses are listed in Table 3 for ease of reading.

**Table 3.** All hypotheses in this study.

| No. | The Content of Hypothesis |
|---|---|
| Hypothesis 1 | CEGE can significantly decrease the CEI of importing countries. |
| Hypothesis 2 | CEGE increases the proportion of clean and renewable energy used or used for power generation in importing countries, thereby reducing its CEI. |
| Hypothesis 3 | The goods in categories 2 and 3 can reduce the CEIIC more effectively than any other category. |
| Hypothesis 4 | CEGE has the strongest inhibitory effect on CEI in South America and Africa, followed by Oceania and Asia, while this effect is insignificant in Europe and North America. |
| Hypothesis 5 | CEGE has a stronger inhibitory effect on the CEI of non-APEC countries than APEC countries. |
| Hypothesis 6 | CEGE has a stronger inhibitory effect on the CEI of SEC countries than the other countries. |

## 3. Methodology

### 3.1. Empirical Model

This paper adopts the Instrumental Variable (IV) approach, one of the three causal identification strategies, to explore the impact of CEGE on the CEIIC. The reasons for choosing the IV approach are as follows. Environment goods trade and CEI may be mutually causal. In addition, despite adding many control variables, there is still the problem of missing variables, which is a "common problem" in econometric models. Therefore, an IV needs to have two characteristics: correlation and exogenous. Specifically, the IV in this paper must be highly correlated with the CEGE and uncorrelated with stochastic errors. Inspired by Frankel and Romer (1999) [9], this paper takes a Two-Stage Least Squares (TSLS) regression model for causal identification [27]. This paper sets the following model to explore the effects of CEGE on the CEIIC.

$$CEI_{it} = \beta_0 + \beta_1 CEGE_{it} + \beta_2 ENERMIX_{it} + \sum_k \alpha_k X_{kit} + \mu_{it} \tag{1}$$

where i denotes country, t is the year when the trade occurred, CEI represents the carbon emission intensity of the importing country, CEGE shows China's export value of environment goods, ENERMIX signifies energy mix, X represents a series of control variables, and $\mu$ denotes stochastic errors.

### 3.2. Core Variables

3.2.1. Carbon Emission Intensity (CEI)

CEI is the explained variable of this study. It is a critical indicator of sustainable development, generally defined as the amount of carbon dioxide emissions per unit of GDP. Due to the unavailability of data at the provincial level, scholars mainly use the carbon emission coefficient of the Intergovernmental Panel on Climate Change [28–30], night light data (DMSP/OLS or NPP/VIIRS) [31–35] to calculate the carbon dioxide emissions. This paper derives carbon dioxide emissions data at the country level directly from the Integrated Carbon Observation System website. Moreover, because this paper aims to explore whether the export of environment goods has an inhibitory effect on CEI, there is no need to consider the issue of actual $CO_2$ emissions calculated on the basis of value-added or final consumption. In other words, it is not necessary to trace the carbon footprint through the world input-output table. Therefore, this paper measures CEI using the carbon dioxide emissions per unit of GDP (constant 2010 US dollar) in each country.

3.2.2. China's Environment Goods Export (CEGE)

In this paper, China's export value of environment goods is a core explanatory variable, measured by the sum of the export values of 54 environment goods under the 6-digit HS code. Due to a probable two-way causal relationship between CEGE and CEI, this paper solves the endogeneity problem through the instrumental variable (IV) method and the TSLS method. See the addressing endogeneity section for details.

3.2.3. Energy Mix (PRECTT and PREPGTT)

The energy mix is an intermediary variable that is used to verify the internal impact mechanism of CEGE on CEI. CEI has a strong correlation with the energy mix because different energy substances have different carbon emission coefficients [36]. For example, the carbon emission factor of coal is greater than that of oil and natural gas, while clean energy does not emit carbon dioxide. As mentioned in the theoretical mechanism section, the CEGE (especially renewable energy goods) may change the energy mix of the importing country, and the energy mix will directly affect the CEI. In this paper, proxies for energy mix are the proportion of renewable energy consumption to total energy consumption (PRECTT) and the proportion of renewable energy power generation to total power generation (PREPGTT).

### 3.3. Control Variables

3.3.1. GDP Per Capita (GDPPC)

The GDP per capita of importing countries can measure their economic development level, which has a strong correlation with the intensity of carbon emissions. A higher GDP per capita means a potentially larger economy and more frequent economic activity. Irrespective of a positive or negative correlation between these two variables, the conclusion will differ in various conditions. When the growth rate of the GDP in a country is lower than the growth rate of $CO_2$ emissions, they are negatively correlated. On the contrary, the two are positively correlated. The original unit of GDP per capita is the current U.S. dollar. This paper uses the Consumer Price Index (CPI) to convert the unit of GDP per capita into the constant 2010 US dollar. This study adds the natural logarithm values of this variable into the model.

3.3.2. Foreign Direct Investment (FDI)

Previous studies have two different views regarding the nexus of FDI and CEI. On the one hand, with the continuous improvement of domestic environmental regulations in the home country, FDI transfers high-polluting industries to host countries, which increases their CEI [37]. On the other hand, FDI reduces CEI since it may bring foreign advanced green production technologies [38–40]. To show which perspective is dominant, this paper considers FDI as a control variable in natural logarithm form.

### 3.3.3. Industrial Structure (INDSTR)

This study argues that CEI could be impacted by the proportion of secondary industries in a country and takes the proportion of the value-added of the secondary industry to the GDP in natural logarithm as a proxy for the industrial structure, following Zhang et al. (2014) [41]. In Zhang et al. (2014) [41], the proportion of the value-added of the tertiary industry to GDP is used for the industrial structure proxy.

### 3.3.4. Urbanization Rate (URB)

This paper adds the urbanization rate of each country in the world as one of the control variables. Urbanization mainly affects carbon emissions in the following ways. Urbanization stimulates the innovation and spread of energy technology, thereby improving energy efficiency and reducing the CEI. Meanwhile, the energy consumption of urban residents is much greater than that of rural residents, which increases the CEI [36,39]. Hence, this paper considers the urbanization rate in the form of a natural logarithm, which is measured by the proportion of the urban population to the total population as a control variable.

### 3.3.5. Trade Openness (TRAO)

This study takes trade openness as a control variable. Indeed, the effect of trade on the environment has been debated. Previous literature proposes three different views regarding this. One perspective shows that trade will have a negative impact on the domestic environment due to the adoption of looser environmental policies in open countries. The other perspective argues that a country can obtain higher income from trade so that the country can purchase more products that are beneficial to the environment, thereby contributing to the enhancement of environmental conditions. The third perspective is the Environmental Kuznets Hypothesis, which comprises the mentioned two perspectives. According to this hypothesis, trade will increase environmental pollution in developing economies since they concentrate on economic aspects and ignore the environment at the early stage. However, trade will decrease environmental pollution in developed countries due to their relatively advanced and energy-efficient technologies [3,42–44]. Ignorant of the direction of the relationship, various studies confirm that trade has a significant relationship with environmental pollution [4–6]. Therefore, when sorting out the influencing factors of the environment (such as the CEI), some scholars take trade openness as one of the control variables. Most scholars directly use the proportion of trade value to GDP to measure trade openness. This paper intends to use the index "Freedom to Trade Internationally" in the Economic Freedom of the World database published by the Fraser Institute to measure the degree of trade openness. The index value ranges from 0 to 10. 0 represents the lowest openness, and 10 represents the highest openness. This study introduces the natural logarithm of this variable.

### 3.4. Addressing Endogeneity

Using the Ordinary Least Squares (OLS) method may have endogeneity problems caused by reverse causality since each country's CEI may affect the import of environment goods. In addition, the problem of missing variables is a "common problem" in the econometric models in existing research. In fact, the reason for the existing endogenous problem boils down to the fact that the key explanatory variable is correlated with stochastic errors, so consistent estimators cannot be obtained, which in turn affects the research conclusions. Scholars mainly alleviate the endogeneity problem using PSM, DID, IV, panel data fixed-effect models, and adding control variables. This paper intends to use the IV method for causal identification. The principle is to decompose the endogenous explanatory variables into the "exogenous part" determined by the instrumental variables and the "rest part (including the endogenous part)" related to the stochastic errors. The "exogenous part" is used to substitute for the original endogenous explanatory variable, and the "rest part" and the original stochastic errors are combined into new stochastic errors. Obviously,

the "exogenous part" is uncorrelated with the stochastic errors, which means that the estimators are consistent in estimating the coefficients of the core explanatory variables.

Inspired by Frankel and Romer (1999) [9] and Frankel and Rose (2002) [8], this paper intends to select geographical factors as the instrumental variables of CEGE. Specifically, the IV includes the population of the importing country, whether the two sides have a common language, whether the importing country is a landlocked country, and other factors of geographical characteristics. These factors are highly correlated with the CEGE and may be able to affect CEI only through trade. Therefore, the above geographical factors meet the selection criteria of IV. Based on the above IV, a regression was performed using TSLS.

### 3.5. Data Sources and Data Processing

Considering the availability of data, this study utilizes the panel data of 195 countries from 2012 to 2020. The carbon emission data at the country level are obtained from the Integrated Carbon Observation System. Trade data are from the UN Comtrade. Also, the World Bank database is the source of GDP per capita, CPI, energy mix, industrial structure, and urbanization rate. The FDI data comes from the UNCTAD database. The trade openness data are from the Economic Freedom of the World database published by the Fraser Institute. The source of the IV data is the CEPII database.

## 4. Results and Discussion

### 4.1. Descriptive Statistics

Table 4 represents the descriptive statistics of each variable in the model.

**Table 4.** Descriptive statistics.

| Variable | Obs | Mean | St.Dev. | Min | Max |
| --- | --- | --- | --- | --- | --- |
| CEI | 1683 | 0.00 | 0.00 | 0.00 | 0.03 |
| CEGE | 1683 | 300,000,000.00 | 949,000,000.00 | 0.00 | 9,210,000,000.00 |
| PRECTT | 1683 | 31.33 | 28.44 | 0.00 | 94.88 |
| PREPGTT | 1683 | 32.35 | 32.56 | 0.00 | 100.00 |
| GDPPC | 1683 | 12,543.16 | 18,091.62 | 18.29 | 113,796.50 |
| FDI | 1683 | 148,689.40 | 580,080.80 | 0.00 | 10,800,000.00 |
| INDSTR | 1683 | 25.58 | 11.42 | 4.56 | 77.31 |
| URB | 1683 | 58.56 | 22.84 | 11.19 | 100.00 |
| TRAO | 1683 | 7.02 | 1.22 | 1.96 | 9.56 |

### 4.2. Benchmark Regression

For the robustness of estimation results, this paper considers the heteroscedasticity problem. In the case of heteroskedasticity, the GMM and iterative estimations are more efficient than TSLS estimation. In Table 5, columns 1, 2, and 3 reveal the estimation results of the TSLS, GMM, and iterative GMM, respectively. Then, this research compares the estimation results of the TSLS, GMM, and IGMM to see if they are similar or dissimilar. The results all imply that CEGE inhibits the CEIIC. Since the variables are in natural logarithmic form, their coefficients are elasticities of the dependent variable [45]. Hence, the results of TSLS estimations show that a 1% increase in the export value of China's environment goods decreases the CEI in each country by 22.61% on average. In addition, trade openness inhibits the CEI, which is statistically insignificant. This result is consistent with the conclusion drawn by previous studies.

**Table 5.** Benchmark regression.

| | (1) | (2) | (3) |
|---|---|---|---|
| | **TSLS** | **GMM** | **IGMM** |
| ln*CEGE* | −0.2261 *** (0.0416) | −0.2119 *** (0.0394) | −0.2113 *** (0.0394) |
| ln*GDPPC* | −0.3831 *** (0.0339) | −0.3752 *** (0.0332) | −0.3748 *** (0.0332) |
| ln*FDI* | 0.2787 *** (0.0537) | 0.2605 *** (0.0511) | 0.2595 *** (0.0511) |
| ln*INDSTR* | 0.7679 *** (0.0608) | 0.7654 *** (0.0592) | 0.7654 *** (0.0592) |
| ln*URB* | 0.2846 *** (0.0816) | 0.2908 *** (0.0795) | 0.2916 *** (0.0794) |
| ln*TRAO* | −0.0849 (0.1536) | −0.0958 (0.1522) | −0.0958 (0.1521) |
| _cons | −6.9018 *** (0.4621) | −7.0204 *** (0.4473) | −7.0279 *** (0.4471) |
| Observations | 1683 | 1683 | 1683 |
| R-squared | 0.0279 | 0.0473 | 0.0482 |
| Number of IV | 3 | 3 | 3 |

Note: Parentheses show robust standard errors; *** indicate statistical significance at 1% levels.

### 4.3. Mechanism Tests

To check Hypotheses 1 and 2, this paper uses two variables, PRECTT and PREPGTT, as proxy variables for energy mix and carries out a mechanism test. The results are as follows.

According to columns (2) and (3) of Table 6, the CEGE increases the PRECTT by 14.67%, which decreases the CEIIC by 43.84%.

**Table 6.** Mechanism tests results.

| | (1) | (2) | (3) | (4) | (5) |
|---|---|---|---|---|---|
| Explained variable | ln*CEI* | ln*PRECTT* | ln*CEI* | ln*PREPGTT* | ln*CEI* |
| ln*CEGE* | −0.2261 *** (0.0416) | 0.1467 *** (0.0194) | −0.1611 *** (0.0342) | 0.1679 *** (0.0252) | −0.2030 *** (0.0401) |
| ln*PRECTT* | | | −0.4384 *** (0.0317) | | |
| ln*PREPGTT* | | | | | −0.1365 *** (0.0198) |
| Control variables | Yes | Yes | Yes | Yes | Yes |
| Observations | 1683 | 1683 | 1683 | 1683 | 1683 |
| R-squared | 0.0279 | 0.2723 | 0.2596 | 0.0573 | 0.0880 |

Note: Parentheses show robust standard errors; *** indicate statistical significance at 1% levels.

In addition, renewable energy has a variety of uses. Solar energy is extremely versatile and can be used in solar water heaters, solar boilers, solar cookers, solar lamps, and power generation. Wind energy can be used to drive machinery and equipment directly and generate electricity. Hydro energy is mainly used for power generation. Another proxy for energy mix is the PREPGTT since the main function of environment goods is the usage of renewable energy (solar, hydro, wind) for generating electricity. Columns (4) and (5) of Table 6 show the regression results. According to the results, the CEGE increases the PREPGTT by 16.79% in importing countries, which is greater than the increment in PRECTT. However, the PRECTT reduces the CEIIC only by 13.65%, which is much lower than that of

the PREPGTT. Thus, both regression results show that CEGE can affect the energy mix in importing countries, thereby reducing the CEIIC.

Moreover, column (1) of Table 6 reveals the total effect of CEGE on the CEIIC. The results show that CEGE reduces the CEIIC by 22.61%, which is significantly larger than the corresponding values shown in columns (3) and (5) of Table 6. This result shows other channels through which the CEGE affects the CEIIC.

*4.4. Mechanism Tests*

4.4.1. Heterogeneity Impact of Different Goods

Based on the different environmental protection functions of goods, this paper divides environment goods into five categories (refer to the mechanisms and hypotheses section for more details), and the regression results are displayed in columns (1) to (5) of Table 7, respectively. According to Table 7, all kinds of environment goods exported by China have significantly reduced the CEIIC. This shows that the CEGE has a good inhibitory effect on the CEI and has effectively played a role in governance and improvement of the ecological environment. Among goods, category 4 impacts the CEIIC most significantly. When the export value of category 4 increases by 1%, the CEIIC decreases by 23.27%. When the export value of the category 5 goods increases by 1%, the CEIIC decreases by 21.69%. Surprisingly, when the export value of the category 3 goods used for the exploitation of clean and renewable energy increases by 1%, the CEIIC will decrease by 19.09%, which shows that its impact is lower than that of categories 4 and 5. In addition, a 1% increase in the export value of categories 1 and 2 reduces the CEIIC by 15.64% and 15.08%, respectively.

**Table 7.** Regression results of different categories of goods.

| Variables | (1) | (2) | (3) | (4) | (5) |
|---|---|---|---|---|---|
| ln*CEGE* | −0.1564 *** (0.0257) | −0.1508 *** (0.0267) | −0.1909 *** (0.0339) | −0.2327 *** (0.0489) | −0.2169 *** (0.0439) |
| Control variables | Yes | Yes | Yes | Yes | Yes |
| Observations | 1683 | 1683 | 1683 | 1683 | 1683 |

Note: Parentheses show robust standard errors; *** indicate statistical significance at 1% levels.

These results have some explanations and interpretations. One explanation is that renewable energy is not yet widely used worldwide. For example, according to the International Energy Agency, only 5% of South Korea's electricity comes from power generation of renewable energy. Therefore, its impact on CEI is still limited, which may be the result of the relatively high cost of renewable energy exploitation. Another explanation is that category 4 is used to dispose of waste and pollutants, and category 5 is used for environmental and pollution monitoring. Obviously, category 4 is instrumental in effectively saving resources and reducing the CEI. Also, category 5 can play a supervisory role and effectively prevent enterprises from using extensive production methods, which will greatly reduce the CEI.

4.4.2. Heterogeneity Impact on Countries of Different Continents

Table 8 represents the regression results. First, the CEGE has reduced the CEI of countries in South America, Africa, Oceania, and Asia. Second, the CEI of South American countries has been impacted by the CEGE most evidently, with a CEI reduction of 51.14%; African countries are close behind, with a CEI reduction of 34.53%. Oceania countries rank third, with a CEI reduction of 27.89%. Asian countries rank fourth, with a CEI reduction of 16.70%. Third, both Europe and North America are insignificantly affected.

**Table 8.** Regression results of different continental countries.

| | (1) | (2) | (3) | (4) | (5) | (6) |
|---|---|---|---|---|---|---|
| | **Asia** | **Africa** | **North America** | **South America** | **Europe** | **Oceania** |
| ln*CEGE* | −0.1670 *** (0.0411) | −0.3453 *** (0.1085) | 0.0034 (0.0259) | −0.5114 *** (0.1110) | −0.0150 (0.0136) | −0.2789 *** (0.1053) |
| Control variables | Yes | Yes | Yes | Yes | Yes | Yes |
| Observations | 396 | 468 | 216 | 99 | 360 | 144 |

Note: Parentheses show robust standard errors; *** indicate statistical significance at 1% levels.

The possible reasons for the differences in the estimated results are as follows: Asia, Africa, South America, and Oceania mainly consist of developing countries, of which green industries are relatively backward, and these countries have imported environment goods from China for a long time. For example, China's "One Belt, One Road" renewable energy international cooperation has been aligned with Africa's energy strategy, and a 100 GW renewable energy market has been formed in Africa. In this way, China can export renewable energy power generation equipment to such countries, thereby changing its energy mix and reducing its CEI. However, North American and European countries are not the main destinations for China exporting environment goods since these countries are more advanced than China in the field of green industries. Therefore, it is not difficult to understand the fact that the CEGE has insignificantly affected the CEI in North American and European countries.

4.4.3. Heterogeneity Impact on APEC and Non-APEC Countries

First, the CEGE reduces the CEI in non-APEC countries by 25.59%, according to column (1) of Table 9. Second, the CEGE has an insignificant impact on the CEI of APEC countries based on column (2) of Table 9.

**Table 9.** Regression results of models 1 and 2 for APEC and non-APEC countries.

| | (1) | (2) |
|---|---|---|
| | **Non-APEC Countries** | **APEC Countries** |
| lnCEGE | −0.2559 *** (0.0464) | −0.0383 (0.0275) |
| Control variables | Yes | Yes |
| Observations | 1521 | 162 |

Note: Parentheses show robust standard errors; *** indicate statistical significance at 1% levels.

The reasons for the different regression results may be as follows. Compared with non-APEC countries, APEC countries pay more attention to environmental products because they have taken the lead in reaching a consensus on the list of environment goods. It means that the industries related to the environment goods of APEC countries are relatively mature, and there is no import dependence on Chinese environment goods. Therefore, the CEGE will not significantly affect the CEI of APEC countries. On the contrary, the industries related to environment goods in non-APEC countries are relatively backward. Therefore, non-APEC countries import environment goods from China in large quantities, which is conducive to reducing these countries' CEI.

4.4.4. Heterogeneity Impact on SEC and Non-SEC Countries

According to column (1) of Table 10, the CEGE significantly reduces the CEI in the SEC and non-SEC countries by 48.95% and 23.32%, respectively. This result implies that the CEGE effect in the SEC countries is twice that in the non-SEC countries.

**Table 10.** Regression results of models 1 and 2 for SEC and non-SEC countries.

|  | (1) | (2) |
| --- | --- | --- |
|  | **Non-SEC Countries** | **SEC Countries** |
| ln*CEGE* | −0.2332 *** | −0.4895 *** |
|  | (0.0436) | (0.0915) |
| Control variables | Yes | Yes |
| Observations | 1494 | 189 |

Note: Parentheses show robust standard errors; *** indicate statistical significance at 1% levels.

This result has the following reasons. The SEC countries are the core areas for the green "Belt and Road" construction. According to the Big Data Analysis 2020, two of only three 100 GW renewable energy projects are in the New Eurasian Land Bridge Economic Corridor and the Bangladesh-China-India-Myanmar Economic Corridor. In addition, the China-Pakistan Economic Corridor and China-Central Asia-West Asia Economic Corridor are important cooperation areas for the exploitation of renewable energy. Hence, it is reasonable that the CEGE has a stronger impact on SEC countries than on non-SEC countries.

*4.5. Robustness Tests*

4.5.1. Identification Test

This paper intends to use the population of the importing country, whether the two sides have a common border, and whether the importing country is a landlocked country as the instrumental variables of the CEGE. The instrumental variables need to satisfy two conditions.

First, all instrumental variables should be exogenous variables, which means that instrumental variables should be unrelated to stochastic errors. In this regard, this paper conducts over-identification and under-identification tests simultaneously. Table 11 represents the test results. The results accept the null hypothesis that all IVs are exogenous variables. The under-identification test shows that the number of endogenous variables is less than or equal to that of IVs, which meets the requirements.

**Table 11.** Robustness test results.

| Identify Category | Statistical Value | $p$ Value |
| --- | --- | --- |
| Over-identification test | 2.04 | 0.3612 |
| Under-identification test (Kleibergen–Paap rk LM) | 122.68 | 0.0000 |
| Weak identification test | 601.39 (C.D. Wald F) | >13.91 (5%) |
|  | 157.60 (K.P. rk Wald F) | 0.0000 |
| Endogenous test | 363.92 (Hausman) | 0.0000 |
|  | 365.44 (DWH) | 0.0000 |
|  | 464.58 (DWH) | 0.0000 |
|  | 39.02 (ENDOG) | 0.0000 |

Second, all instrumental variables should be related to endogenous explanatory variables. In this regard, this paper uses three methods. The first method is to check whether the IVs coefficient is significant in the first-stage regression of TSLS. The second is to perform a less sensitive Limited Information Maximum Likelihood (LIML) estimation. The principle of this method is to compare the regression results of the TSLS with those of the LIML. If these results are close, there is no weak identification problem. The third method is the Wald test. This paper represents the results of only the third method (the results of the first and second methods are available upon request). Table 11 offers the results of the third method. These results show no weak identification problem. In addition, this paper conducts an IV redundancy test. The results show that all instrumental variables are not

redundant variables. Therefore, it is reasonable to use these three variables as instrumental variables.

### 4.5.2. Endogenous Test

The premise of using the IV method is that the core explanatory variable is an endogenous variable, requiring an endogeneity test. This paper uses three methods for this test. (1) The first one is the Hausman test. On the basis of Table 11, the test results reject the null hypothesis that the core explanatory variable is an exogenous variable. However, the Hausman test needs to satisfy the precondition of homoscedasticity. Therefore, it is necessary to relax the assumption and conduct further testing. (2) The second test is the heteroskedasticity robust DWH test. With regard to Table 11, the DWH test has two statistics, and all the results show that the core explanatory variable is not exogenous. (3) The third test is the endog command. Based on Table 11, the results show that the core explanatory variable is endogenous. Thus, the results of all the tests show that the core explanatory variable, CEGE, is an endogenous explanatory variable, and the IV method can be used for causal identification.

### 4.5.3. Year Fixed Effect

The heterogeneity impact at the country level has been examined above, which means the impact of individual differences on regression has been considered. This paper will further add the year-fixed effect to represent the impact of these variables that change only with time, such as exchange rates, on the CEI. According to Table 12, after adding the year-fixed effect, there is no significant difference between the regression results and the benchmark regression results.

**Table 12.** Regression results after adding the year-fixed effect.

| Variable | (1) | (2) |
|:---:|:---:|:---:|
| ln*CEGE* | −0.2261 *** (0.0416) | −0.2233 *** (0.0415) |
| Control variables | Yes | Yes |
| Year fixed effect | No | Yes |
| Observations | 1683 | 1683 |

Note: Parentheses show robust standard errors; *** indicate statistical significance at 1% levels.

## 5. Conclusions and Policy Implications

### 5.1. Main Research Conclusions

This paper empirically explores the effects of the CEGE on the CEIIC and obtains the following conclusions.

The benchmark regression results show that the CEGE significantly inhibits the CEIIC. Mechanism tests show that the CEGE has significantly increased the energy mix (PRECTT and PREPGTT), thereby reducing the CEIIC.

Heterogeneity analysis shows that, at the product level, the export of the goods used for the disposal and recycling of waste and pollutants impacts the CEIIC most seriously, followed by goods used for environmental and pollution monitoring. Surprisingly, the impact of the goods used for the exploitation of clean and renewable energy on the CEIIC is weaker than the first two goods. In addition, bamboo spliced floors and biomass-fired machines and equipment can significantly reduce the CEIIC.

At the country level, the CEGE has the greatest effect on the CEI in South America, followed by Africa, Oceania, and Asia, but an insignificant effect on the CEI in European and North American countries, consistent with Hypothesis 4. The CEGE has a stronger inhibitory effect on the CEI of non-APEC countries than APEC countries, which may be a result of the fact that APEC countries have relatively mature industries of environment goods. The inhibitory effect of the CEGE on the CEI of SEC countries is much greater

than that of non-SEC countries since most Chinese energy companies are concentrated in the SEC area, which is also consistent with the discussion in the Big Data Analysis 2020, which shows that SEC is the key area for overseas cooperation in China's renewable energy projects.

### 5.2. Policy Implications

#### 5.2.1. Vigorously Promote Trade in Environment Goods to Support Global Climate Action

The CEGE can significantly affect the CEIIC. Therefore, in order to contribute to global climate governance, China should vigorously promote trade with other countries in the field of environment goods. Also, China should promote the signing of trade liberalization agreements for environment goods with more countries for a stronger contribution to global climate action. In addition, intra-industry trade is also important since it helps enhance the trade competitiveness of countries, thereby improving the development speed and performance of the product, which is conducive to achieving global carbon reduction more efficiently.

#### 5.2.2. Strengthen the Trade of Renewable Energy Goods and Promote the Gradual Replacement of Traditional Fossil Energy with Renewable Energy

In view of the significant role of the energy mix in reducing the CEI, this paper argues that clean and renewable energy is of great significance in global climate governance. At present, China has a comparative advantage in the field of renewable energy and should further expand its export of renewable energy goods to developing countries. In addition, implementing renewable energy projects is the most key and direct way to realize the transformation of the energy mix. Therefore, it is necessary to accelerate the implementation of the signed projects and sign more energy projects, by which the application scope of renewable energy will be extended.

#### 5.2.3. Increase Technical and Equipment Support for Developing Countries and Seek Technical Cooperation with Developed Countries in Europe and America

At present, the CEGE has a strong inhibitory effect on the CEI of South America, Africa, Oceania, and Asian countries. Therefore, it is recommended that China continue to increase the export of environment goods to the countries from the above continents and seek more cooperation in energy projects. China should also provide maximum support to developing countries in terms of capital, technology, and equipment to help them realize energy transformation and reduce the CEI. Moreover, China should seek deep technical cooperation and intra-industry trade with the European and North American countries in the field of environment goods, which can substantially contribute to improving the technical level of environment goods, thereby improving the efficiency of the CEI reduction.

#### 5.2.4. Expand the Scope of Trade Liberalization Environment Goods through the United Nations, WTO, APEC, and Other Dialogue Mechanisms

As environmentally friendly products, the environment goods do not cause damage to the environment and are also used to protect the environment. Therefore, it is urgent to expand the scope of the "APEC LIST OF ENVIRONMENT GOODS" based on the comparative advantages of each country. More such goods should be promoted to achieve trade liberalization in order to allocate environment goods resources worldwide effectively and serve global climate action. China should play a promotion role under the high-level dialogue mechanisms of the United Nations, WTO, and APEC so as to contribute to the global carbon reduction work. Moreover, the signatories of the APEC list of environment goods should not be limited to APEC countries, and it is feasible to encourage more countries to jointly sign the "APEC+ LIST OF ENVIRONMENT GOODS".

### 5.2.5. Expand the Market Scope and Business Scope of Chinese Energy Companies "Going Out" through the Advantages of "One Belt, One Road"

At present, the key area of China's renewable energy cooperation is the SEC area, which is mainly concentrated in the fields of photovoltaic, hydro, and wind power generation, but it still has great development potential. Specifically, first, the market scope of renewable energy cooperation still has the potential to increase. In terms of individual regions, the number of renewable energy projects currently signed has yet to be increased. In terms of the number of countries, the scope of countries that are currently conducting energy cooperation with China needs to be further expanded. Second, policymakers should promote the business scope of renewable energy cooperation, for example, cooperation in sewage treatment, solid waste treatment, waste incineration power generation, biomass energy, and others.

**Author Contributions:** Conceptualization, W.H.; methodology, X.C.; software, X.C.; validation, W.H., X.C. and Y.Z.; formal analysis, X.C.; investigation, Y.Z.; resources, W.H.; data curation, X.C.; writing—original draft preparation, X.C.; writing—review and editing, X.C. and Y.Z.; visualization, X.C.; supervision, W.H.; project administration, W.H.; funding acquisition, W.H. All authors have read and agreed to the published version of the manuscript.

**Funding:** This work receives funding from the Liaoning Revitalization Talent Program of the Liaoning Provincial Government [grant numbers: XLYC2002116].

**Institutional Review Board Statement:** Not applicable.

**Informed Consent Statement:** Not applicable.

**Data Availability Statement:** The data used in this article is based on publicly available data from the World Bank and UN Comtrade.

**Conflicts of Interest:** The authors declare no conflict of interest.

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
