# Peer review of "Export of Environment Goods from China, Importing Countries’ Energy Mix, and Carbon Emission Intensity"

_sustainability, doi:10.3390/su16020556_

Round 1

Reviewer 1 Report

Comments and Suggestions for Authors

Manuscript Comments:

Abstract. The content of the abstract should include quantitative data that assists in the comprehension of the qualitative descriptions provided and the findings presented.

Introduction. Here, I highlight two relevant points that need to be reviewed: (i) I consider the introduction to be excessively lengthy as it includes topics that could well be placed in the methodology section or even in the discussion of results, hence I suggest making the presented information more concise; (ii) a purpose of an introduction is to incorporate pertinent data about the context of the study subject, as this is where the interest in and relevance of its study originates. Therefore, there is a noticeable absence of relevant data that would indicate the importance of the subject matter, both globally and locally; and (iii) It is necessary to include specific background information on studies that have addressed similar situations to this study, providing more insight into what has already been accomplished in this area.

Hypothesis 2. In this case, it should be clarified that there is a limitation; if renewable energy is not generated using efficient equipment and appropriate controls, collateral pollutants that are counterproductive can be produced. It is important that each hypothesis also mentions the potential limitations of these assumptions.

Lines 225-227, 248-250. Even in the case of renewable energy, it is necessary to consider the complete life cycle to understand the actual impact of GHG emissions. Furthermore, only these emissions are relevant for the impact of installations of such magnitude that tend to modify ecosystems. Therefore, it is again important to mention the limitations of the proposed approach.

Line 260. Citations of the background for the included equations must be incorporated, regardless of their originality. There should be references from where the inspiration was drawn.

Income Level per Capita. It is not clear whether the GDP per capita, which takes into account purchasing power, is used; this measure is most appropriate for assessing these types of economic variables.

Lines 310-313. This statement needs to be substantiated with a substantial amount of consulted literature, meaning not just one or two sources, but many more. Without this, the assertion remains weak.

Lines 318-321. Similar to the previous comment, the assertions made throughout the methodology must be substantiated.

Data Sources and Data Processing. These data should be included in the appendix to ensure the accuracy of the information used.

Results and Discussion. Throughout this section, the analysis of results in comparison with other research is very limited. This part is crucial for emphasizing the results obtained in the context of other studies. Without this, the results presented are less credible.

Beyond the statistical variables used to demonstrate the correlation of the parameters employed, it is also important to include both absolute and relative trends of the values handled. For instance, the relationship between GDP per capita and the use of renewable energy, to cite an example. This would enrich the findings of the current study.

Lines 562-567. The content described in this paragraph appears to be a significant finding of the manuscript, therefore, it requires more substantial argumentation.

Conclusions. The content of the conclusions leans more towards a series of recommendations rather than an analysis of the findings and their scope in light of the limitations that could have affected the results. It is advisable to add an analysis in this regard."

Comments on the Quality of English Language

A comprehensive review of the English is recommended to improve the use of phrases and words, which in some cases makes it difficult to understand the text.

Reviewer 2 Report

Comments and Suggestions for Authors

Dear Sir

Thank you for inviting me to review this paper. I think that this paper is good. The methodology is good. The results are diversified and all can support for the objectives that are discussed in the introduction.

I have some comment for improving this paper:

1/ All hypotheses should be shown in the Chart. This can be supported for the readers.

2/ The Descriptive statistics should be shown the original data instead of the log.

3/ R-squared is too low. Why? This is the weakest point in this study. Please comment on this result. (See Table 4)

4/ Please show the number of groups, the number of instruments of GMM estimates.

5/ Please explain why Year fixed effect should be discussed?

Thank you

Comments on the Quality of English Language

 Minor revision

Reviewer 3 Report

Comments and Suggestions for Authors

The topic proposed by the authors, even though not new, is very challenging, because, as they also mention, the world is facing a rising global temperature and China is the largest annual carbon emitter. Therefore, their purpose is clearly stated both in the abstract and in the introductory part: “to empirically explore the impact of China's Environment Goods Export on the Carbon Emission Intensity of Importing Countries”.

Even though the theoretical background is very well contextualized, the authors presenting in a clear manner the previous studies regarding the effects of trade on environment, they should have included in the paper a distinct part for the “literature review” (based on which to develop the research hypotheses). 

The methodology is appropriate for this research and it is very well presented. It clearly states the used model, explains the variables and presents the data sources. 

The arguments they offer in the results and discussion part are convincing.

The conclusions are consistent with the evidence and arguments presented and they address the main objective of the paper.

Comments on the Quality of English Language

English language is fine, with minor errors.

Round 2

Reviewer 1 Report

Comments and Suggestions for Authors

Dear Authors,

The suggested changes have been implemented in the article with adequate technical and argumentative proficiency. Therefore, I consider that the manuscript is ready for publication.

Comments on the Quality of English Language

Regarding the English composition, I suggest that the authors conduct a thorough final review to identify any possible errors.

Reviewer 2 Report

Comments and Suggestions for Authors

Dear Sir

I agreed with this version. However, I think that the language should be more smooth.

Thank you

Comments on the Quality of English Language

Dear Sir

I agreed with this version. However, I think that the language should be more smooth.

Thank you